

# Phylogenetic synthesis of morphological and molecular data reveals insights on the classification of diogenid hermit crabs (Crustacea: Decapoda: Anomura)

Jiao Cheng[1,2,3,*], Wenjie Li[4,*], Yanrong Wang[1,2,3] and Zhongli Sha[1,2,3,4]

[1] Laboratory of Marine Organism Taxonomy and Phylogeny, Qingdao Key Laboratory of Marine Biodiversity and Conservation, Institute of Oceanology, Chinese Academy of Sciences, Qingdao, Shandong, China
[2] Laboratory for Marine Biology and Biotechnology, Qingdao Marine Science and Technology Center, Qingdao, Shandong, China
[3] Shandong Province Key Laboratory of Experimental Marine Biology, Institute of Oceanology, Chinese Academy of Sciences, Qingdao, Shandong, China
[4] University of Chinese Academy of Sciences, Beijing, China
* These authors contributed equally to this work.

Corresponding author
Zhongli Sha, shazl@qdio.ac.cn

## ABSTRACT

The family Diogenidae Ortmann, 1892 is a diverse and abundance group of hermit crabs, but their systematics and phylogenetic relationships are highly complex and unresolved. Herein, we gathered nucleotide sequence data from two mitochondrial (16S rRNA and COI) and two nuclear (NaK and PEPCK) genes for a total of 2,308 bp in length across 38 species from six extant diogenid genera. Molecular data were combined with 41 morphological characters to estimate the largest phylogeny of diogenid hermit crabs to date with the aim of testing the proposed taxonomic scheme of Diogenidae and addressing intergeneric relationships within this family. Despite conflicts between mitochondrial and nuclear DNA trees, the combined-data tree reflects the contributions of each dataset, and improves tree resolution and support for internal nodes. Contrary to traditional classification, our total evidence revealed a paraphyletic Diogenidae based on internally nested representatives of Coenobitidae Dana, 1851. Within Diogenidae, the studied diogenid hermit crabs were split between two clades with high support, which contradicts recent morphological classification scheme for Diogenidae *sensu lato* based on fossil records. The genus *Diogenes* Dana, 1851 was found nested inside *Paguristes* Dana, 1851, which formed a clade being separated from the remainder, pointing towards paraphyly in *Paguristes*. In another clade, *Dardanus* Paulson, 1875 occupied a basal position relative to the other diogenids, while *Calcinus* Dana, 1851 and *Clibanarius* Dana, 1852 showed sister relationships and formed a cluster with *Ciliopagurus* Forest, 1995. Among the morphological characters examined, carapace shield and telson were identified as phylogenetically significant for grouping diogenid genera, while phylogenetic insignificance of gill number was evidenced by its mosaic pattern in diogenid phylogeny. The present study sheds light on the controversial generic phylogeny of Diogenidae and highlights the necessity for thorough taxonomic revisions of this family as well as some genera (*e.g.*, *Paguristes*) to reconcile current classifications with phylogenetic relationships.

## INTRODUCTION

The diogenid hermit crabs (family Diogenidae *sensu lato*) are commonly known as the "left-handed" hermits because most members are characterized by an enlarged left cheliped (*Mclaughlin et al., 2007*). They are found in a wide variety of habitats, such as freshwater, intertidal mangrove swamps, coral reefs and offshore deep water, and are globally distributed throughout a vertical range from the intertidal areas to the deep sea (>1,000 m). As well known for being omnivorous filter feeders, some hermit crabs represent an important group within the intertidal community and in the benthic sublittoral habitat, where they play a critical role in the trophic chain (*Fransozo et al., 1998*). Besides their ecological value, hermit crabs occupy a key position in the decapod evolution, forming a monophyletic group closely related to Brachyura (*Luque et al., 2019*; *Wolfe et al., 2019*). Thus, an improved understanding of this group would shed light on not only its diversity and ecology but also an evolutionary history of decapod crustaceans.

The monophyletic status of Diogenidae has been the subject of controversy. Diogenid hermit crabs have long been considered to constitute a single, monophyletic family Diogenidae (*MacDonald, Pike & Williamson, 1957*; *McLaughlin, 1983*; *Forest, 1984*, *1995*; *Cunningham, Blackstone & Buss, 1992*; *McLaughlin & Lemaitre, 1997*; *Forest et al., 2000*; *McLaughlin, Lemaitre & Sorhannus, 2007*; *McLaughlin et al., 2010*; *De Grave et al., 2009*). However, emerging evidence from adult morphology, spermatozoa and spermatophore morphology as well as molecular data poses challenges to the monophyletic origin of diogenids (*Richter & Scholtz, 1994*; *Tudge, 1997*; *Schnabel, Ahyong & Maas, 2011*; *Tsang et al., 2008*, *2011*; *Bracken-Grissom et al., 2013*; *Landschoff & Gouws, 2018*; *Craig & Felder, 2021*). Notably, no matter mono-, poly- or paraphyletic status recovered for Diogenidae, the terrestrial hermit crab family, Coenobitidae, is found closely allied with Diogenidae in any case, corroborating previous arguments for morphological affinity between diogenids and coenobitids (*McLaughlin, 1983*; *Tudge, 1991*, *1992*; *Richter & Scholtz, 1994*). Recently, *Fraaije (2014)*, based largely on parallel intragastric grooves to the cervical groove, established a fossil diogenid family, Annuntidiogenidae Fraaije, 2014. Based on intragastric groove patterns on the cephalothoracic shield of fossil hermit crabs, *Fraaije, Van Bakel & Jagt (2017)* erected the family, Calcinidae Fraaije, Van Bakel & Jagt, 2017, and subdivided diogenid hermit crabs into three families, Annuntidiogenidae (four genera), Calcinidae (seven genera) and Diogenidae *sensu stricto* (11 genera). However, this classification is mainly based on fossil records, and until cephalothoracic groove patterns, numerous other morphological characters as well as molecular data are fully evaluated across extant diogenid groups, we tentatively retain Diogenidae *sensu lato* for the studied diogenid hermit crabs in the present study.

In addition to arguments for the monophyly of Diogenidae *sensu lato*, the elucidation of internal relationships among diogenid genera using modern methods is a dynamic and continuous debate. Due to the close association between members of the families Coenobitidae and Diogenidae *sensu lato*, recent molecular studies proposed questions

concerning the phylogenetic relationships among diogenid genera (*Calcinus*, *Clibanarius* and *Dardanus*) and coenobitid genera (*Coenobita* Latreille, 1829 and *Birgus* Leach, 1816) (Fig. S1). For instance, *Mantelatto et al. (2006)* traced genetic relationships among selected anomuran decapods based on partial mitochondrial 16S rRNA sequences, placing *Coenobita* as more closely related taxa to *Dardanus* than to *Calcinus* and *Clibanarius*. This finding was subsequently supported by *Bracken-Grissom et al. (2013)* based on two mitochondrial (16S and 12S rRNA) and three nuclear (H3, 18S and 28S rRNA) genes. However, *Coenobita* was found much more related to *Clibanarius* than to *Calcinus* and *Dardanus* based on five nuclear protein-coding genes (*Tsang et al., 2011*). More recently, molecular phylogeny of selected paguroid species from the western Atlantic (*Craig & Felder, 2021*) did not support previous hypotheses of relationships, and instead suggested close relationships between *Calcinus*, *Clibanarius* and coenobitids, placing *Dardanus* in another diogenid clade. It is worth noting that many of the phylogenetic studies above included limited diogenid taxa and produced low support values, and the intergeneric relationships remain poorly resolved, highlighting the need for continued phylogenetic study of this group.

Given all the uncertainties mentioned above, we present a phylogenetic framework of Diogenidae *sensu lato* by integrating morphological characters with newly acquired and previously published sequence data of 38 species from six diogenid genera to evaluate the proposed classification scheme of Diogenidae *sensu lato* (*Fraaije, Van Bakel & Jagt, 2017*) as well as elucidate the phylogenetic relationships among diogenid genera. By examining morphological characteristics of studied diogenid genera, we also endeavour to evaluate the phylogenetical significance of morphological characters of carapace shield, telson and gill number that are currently applied in diogenid species taxonomy. These findings are expected to provide insights into the intergeneric classification of this important anomuran group.

## MATERIALS AND METHODS

### Taxon sampling and data collection

The present study included 38 species from six extant diogenid genera and three outgroup taxa from the families Paguridae Latreille, 1802 (*Pagurus ochotensis* and *P. bernhardus*) and Lithodidae Samouelle, 1819 (*Paralomis dofleini*). A total of 107 nucleotide sequences from 28 diogenid specimens were new to this study, while sequences for all four genes (16S rRNA, COI, NaK and PEPCK) from 13 taxa (including outgroups) were obtained from GenBank and our previous study (*Li et al., 2019*). In light of the importance of coenobitids in diogenid phylogeny (*McLaughlin, 1983*; *Tudge, 1991*, *1992*; *Richter & Scholtz, 1994*), sequences for all four genes of two coenobitid species (*Coenobita rugosus* and *C. violascens*) were also included in this study. Newly included specimens were identified based on morphological characters including carapace shield, cheliped, pleopod, rostrum, stridulating apparatus, gill number and telson. The sampling information of specimens is provided in Table 1. Specimens were preserved in 95% ethanol before laboratory analysis. A total of 168 nucleotide sequences from 43 species were used in this study, with nucleotide sequences of PEPCK gene unavailable for four *Diogenes* species.

The morphological data matrix consisted of 41 characters and 43 species, including 38 diogenid species, three outgroups and two coenobitid species (see Table S1). Codings for morphological characters were scored based on individual examination of studied taxa (Table 1) supplemented by literatures (*McLaughlin, 1974*; *2003*; *Mclaughlin et al., 2007*; *McLaughlin, Lemaitre & Sorhannus, 2007*; *Sha, Xiao & Wang, 2015*). The inapplicable or missing character states were scored as unknown (indicated by '–').

## DNA extraction, PCR amplification and sequencing

Total genomic DNA was extracted from the pereopod or cheliped (5–20 mg) using a DNeasy Blood and Tissue Kit (Qiagen, Hilden, Germany) according to the manufacturer protocol. The total DNA was eluted in 100 uL of sterile distilled $H_2O$ (RNase free) and stored in −20 °C freezer. The partial sequences of two mitochondrial genes were amplified by polymerase chain reaction (PCR) using the universal primers: dgLCO1490/dgHCO2198 for COI (*Meyer, 2003*) and 16S1471/16S1472 for 16S rRNA (*Crandall & Fitzpatrick, 1996*). The two nuclear genes were amplified using the following primers: NaK for-a/NaK rev3, NaK for-b/NaK rev3, NaK for-b/NaK rev, NaK for-b/NaK rev2 for NaK (*Tsang et al., 2008*, *2014*) and PEPCK for/PEPCK rev, PEPCK for/PEPCK rev3, PEPCK for2/PEPCK rev, PEPCK for2/PEPCK rev3 for PEPCK (*Tsang et al., 2008*). The primer information was listed in Table S2. The PCR amplification was conducted in 25 uL reaction mixture, containing 1–2 uL of extracted DNA, 1uL of Taq polymerase, 2.5 uL PCR buffer, 0.5 uL dNTPs, 0.5 uL of each primer and 10–12 uL $H_2O$. The PCR cycling conditions included an initial denaturation for 3 min at 94 °C, followed by 30 cycles of 30 s at 94 °C, 30 s at 48–53 °C depending on gene fragments and specimens, 1 min at 72 °C and a final extension at 72 °C for 10 min. The PCR products were purified using the QIA-quick gel purification kit (Qiagen, Hilden, Germany). Purified PCR products were bidirectionally sequenced using the same primer pairs for PCR amplification by ABI 3730xl DNA Analyzer (Applied Biosystems, Foster City, CA, USA). All sequences obtained in the present study were deposited in GenBank under accession numbers shown in Table 1.

## Phylogenetic analyses

Forward and reverse sequences were cleaned and assembled using SeqMan implemented in the DNASTAR Lasergene software package. Sequences of 16S rRNA were aligned and trimmed by using the program MAFFT ver 7.311 (*Katoh & Standley, 2013*). The protein-coding genes (COI, NaK and PEPCK) were aligned *via* MUSCLE program (*Edgar, 2004*) implemented in MEGA 6.0 (*Tamura et al., 2013*) with default settings. Alignments of these three genes were also verified by translating them into amino acid sequences to check for indels (insertions/deletions) and stop codons. The mitochondrial and nuclear DNA (mtDNA and nDNA) data were initially analyzed individually and then combined to investigate an overall pattern of phylogenetic relationships among diogenid species. Prior to analyses, the concatenated molecular dataset was partitioned for genes, with three protein-coding genes further divided into three codon positions (10 partitions in total). The best partitioning strategy and best fitting substitution models for each partition were then selected in PartitionFinder 2.1.1 according to the Bayesian information

**Table 1 List of diogenid species used in this study.**

| Family | Genus | Species | Sampling locality | GenBank accession number | | | |
|---|---|---|---|---|---|---|---|
| | | | | 16S rRNA | COI | NaK | PEPCK |
| Lithodidae | *Paralomis* | *Paralomis dofleini* | Unknown | HM020962 | HM020912 | GU383059 | GU383026 |
| Paguridae | *Pagurus* | *Pagurus bernhardus* | Unknown | MH917117 | MG935018 | GU383056 | GU383022 |
| | | *Pagurus ochotensis* | Yellow China Sea | MH508055 | MH508058 | MH508076 | MH508087 |
| Diogenidae | *Calcinus* | *Calcinus elegans* | Fulong, East China Sea | MK610008 | MK610039 | MK747800 | MK747818 |
| | | *Calcinus morgani* | Ximao, South China Sea | MK610009 | MK610040 | MK747790 | MK747819 |
| | | *Calcinus gaimardii* | Hainan, South China Sea | MK610010 | MK610041 | MK747801 | MK747820 |
| | | *Calcinus latens* | Hainan, South China Sea | MK610011 | MK747765 | MK747802 | MK747821 |
| | | *Calcinus laevimanus* | Sanya, South China Sea | MK610012 | MK747766 | MK747803 | GU383005 |
| | | *Calcinus vachoni* | Kenting, East China Sea | MK610013 | MK747767 | MK747804 | MK747822 |
| | | *Calcinus guamensis* | Kenting, East China Sea | MK610014 | MK747768 | MK747805 | MK747823 |
| | | *Calcinus minutus* | Taidong, East China Sea | MK610015 | MK747769 | MK747806 | MK747824 |
| | *Clibanarius* | *Clibanarius longitarsus* | Wenchang, South China Sea | MH508050 | MH508061 | MH508073 | MH508085 |
| | | *Clibanarius corallinus* | Foshan, South China Sea | MH508051 | MH508060 | MH508075 | MH508078 |
| | | *Clibanarius virescens* | Sanya, South China Sea | MH508049 | MH508064 | MH508067 | MH508080 |
| | | *Clibanarius humilis* | Fulong, East China Sea | MH508052 | MH508066 | GU383054 | MH508083 |
| | | *Clibanarius englaucus* | Maoao, East China Sea | MH508046 | MH508057 | MH508074 | MH508079 |
| | | *Clibanarius merguiensis* | Malaysia | MH508047 | MH508059 | MH508072 | MH508084 |
| | | *Clibanarius eurysternus* | Taitung, East China Sea | MH508053 | MH508056 | MH508068 | MH508077 |
| | | *Clibanarius snelliusi* | Danzhou, South China Sea | MH508054 | MH508062 | MH508071 | MH508086 |
| | | *Clibanarius infraspinatus* | Ledong, South China Sea | MH508045 | MH508065 | MH508070 | MH508081 |
| | | *Clibanarius rutilus* | Wenchang, South China Sea | MH508048 | MH508063 | MH508069 | MH508082 |
| | *Ciliopagurus* | *Ciliopagurus strigatus* | Keelung, East China Sea | MK610016 | MK747770 | MW413814 | MW413815 |
| | *Dardanus* | *Dardanus lagopodes* | Sanya, South China Sea | MK610017 | MK747771 | MK747791 | MK747825 |
| | | *Dardanus crassimanus* | Keelung, East China Sea | MK610018 | MK747772 | MK747807 | MK747826 |
| | | *Dardanus setifer* | Sanya, South China Sea | MK610019 | MK747773 | MK747808 | MK747827 |
| | | *Dardanus guttatus* | Lingshui, South China Sea | MK610020 | MK747774 | MK747809 | MK747828 |
| | | *Dardanus hessii* | Sanya, South China Sea | MK610021 | MK747775 | MK747810 | MK747829 |
| | | *Dardanus impressus* | Yilan, East China Sea | MK610022 | MK747776 | MK747811 | MK747830 |
| | | *Dardanus gemmatus* | Sanya, South China Sea | MK610023 | MK747777 | MK747792 | MK747831 |
| | | *Dardanus deformis* | Lingshui, South China Sea | MK610024 | MK747778 | MK747793 | MK747832 |
| | | *Dardanus arrosor* | Hainan, South China Sea | MK610025 | MK747779 | MK747794 | MK747833 |
| | *Diogenes* | *Diogenes avarus* | Changhua, East China Sea | MK610026 | MK747780 | MK747812 | MK747834 |
| | | *Diogenes rectimanus* | Guangxi, South China Sea | MK610028 | MK747782 | MK747795 | N/A |
| | | *Diogenes nitidimanus* | Changhua, East China Sea | MK610029 | MK747783 | MK747796 | N/A |
| | | *Diogenes edwardsii* | Guangxi, South China Sea | MK610030 | MK747784 | MK747797 | N/A |
| | | *Diogenes goniochirus* | Guangdong, South China Sea | MK610031 | MW411962 | MK747798 | N/A |
| | *Paguristes* | *Paguristes doederleini* | Taiwan, East China Sea | MK610032 | MW411963 | MK747814 | MK747835 |
| | | *Paguristes miyakei* | Yilan, East China Sea | MK610033 | MK747785 | MK747815 | MK747836 |
| | | *Paguristes albimaculatus* | Taiwan, East China Sea | MK610034 | MK747786 | MK747799 | MK747837 |
| | | *Paguristes calvus* | Yilan, East China Sea | MK610035 | MK747787 | MK747816 | MK747838 |

(*Continued*)

| Table 1 (continued) | | | | | | | |
|---|---|---|---|---|---|---|---|
| Family | Genus | Species | Sampling locality | GenBank accession number | | | |
| | | | | 16S rRNA | COI | NaK | PEPCK |
| Coenobitidae | Coenobita | *Paguristes seminudus* | Kaohsiung, East China Sea | **MK610036** | **MK747788** | **MK747817** | **MK747839** |
| | | *Coenobita rugosus* | Unknown | KY352235 | MH482050 | EU427116 | EU427185 |
| | | *Coenobita violascens* | Unknown | KJ132519 | AB998664 | EU427115 | EU427184 |

Note:
GenBank accession numbers for sequences are shown. Sequences highlighted in bold are obtained from our study. The letter "N/A" indicates missing sequence for this particular fragment.

criterion (*Lanfear et al., 2017*). The final molecular dataset consisted of nucleotide sequences from the above four genes with a length of 2,308 bp, while the combined dataset included the molecular dataset plus an additional 41 morphological characters.

The maximum likelihood (ML) analysis was implemented with the IQ-TREE web server (*Trifinopoulos et al., 2016*) with the best-fit partition schemes and models (Table S3). Confidence in the resulting topology was assessed using 5,000 ultrafast bootstrap replicates (*Minh, Nguyen & von Haeseler, 2013*) and bootstrap (bs) values >50% are presented on the resulting phylogeny. The Bayesian inference (BI) was conducted using MrBayes v3.2.6 (*Ronquist et al., 2012*) for the molecular and combined (molecular + morphology) datasets, respectively. The Markov k (*Lewis, 2001*) model was used for the morphological characters with equal state frequencies, combined with gamma distributed rates across sites. The partitioning schemes and the nucleotide substitution models determined by PartitionFinder 2.1.1 were applied to our molecular dataset. The 'unlink' option was employed to unlink model parameters across character partitions. Two runs of 1,000,000 generations (Markov chain Monte Carlo, MCMC) were performed, with a sampling frequency of 100 generations. To ensure the convergence of the analyses, we monitored the average standard deviations of split frequencies between two simultaneous runs (<0.01) and the potential scale reduction factor (PSRF, close to 1.0). After discarding burn-in trees sampled prior to stationarity, the 50% majority rule consensus tree was conducted from the remaining sampled trees to estimate Bayesian posterior probabilities (Pp) for each node and parameter estimates for both independent runs. All likelihood values and scored means and variances of posterior were further graphically monitored by Tracer v1.7.1 (*Rambaut et al., 2018*). Pp values > 0.5 are presented on the BI topology. For comparison, phylogenetic topologies based on mtDNA or nDNA sequences were investigated separately using the methods described above. In this study, high support is defined as ≥ 95/70 Pp/bs, moderate support ≥ 85/65 Pp/bs and low support ≤ 84/64 Pp/bs.

## Ancestral character state reconstruction

The morphological characters of carapace shield, telson and gill number are key characters previously used for species diagnosis within Diogenidae *sensu lato* (*e.g.*, *McLaughlin & Provenzano, 1975*; *Rahayu, 2005*; *Sha, Xiao & Wang, 2015*). However, their values in generic diagnosis were unexplored or questionable (*Craig & Felder, 2021*). Ancestral State Reconstruction (ASR) methods were implemented in Mesquite v3.61 (*Maddison &*

*Maddison, 2019*) to examine character evolution of these three morphological characters across studied diogenid genera. Morphological scoring was performed for all studied diogenid hermit crabs and two outgroup species of Paguridae, since the absence of *Paralomis dofleini* (Lithodidae) did not change the intergeneric relationships within Diogenidae *sensu lato* (Fig. S2). Carapace shield: unarmed or with scattered setae = 0, with few scattered spines = 1, densely spinose = 2. Posterior margin of telson: armed = 0, unarmed = 1. Gill number: <13 pairs = 0, 13 pairs = 1, 14 pairs = 2. Additionally, missing data of the species was indicated as "?". These morphological character states were mapped on the topology generated from the combined Bayesian phylogeny. Ancestral states were reconstructed under the likelihood model (Markov k-state one parameter model) and maximum parsimony using the Trace Character History option.

## RESULTS

No matter mtDNA, nDNA or combined molecular dataset was used, the molecular phylogenetic trees derived from ML and BI analyses were congruent. Here, only the ML tree was presented with support values for both ML and BI analyses. A discrepancy between mtDNA and nDNA phylogenies was found, particularly with respect to the positions of the genera *Clibanarius*, *Dardanus* and *Diogenes*, but these discordant nodes were not strongly supported (Fig. S3). The combined molecular tree contained a mixture of clades favoured by the separate mtDNA and nDNA datasets. Specifically, the *Calcinus-Clibanarius-Ciliopagurus-Coenobita* clustering was resolved in favour of mtDNA data, while the *Paguristes-Diogenes* clade was resolved in favour of nDNA data. The genus *Paguristes* was recovered as monophyletic in the mtDNA tree, however, it was found paraphyletic in nDNA and combined-data trees.

The generic relationships revealed by the combined molecular-only phylogeny (Fig. 1) parallel the patterns seen in the combined (molecular + morphology) phylogeny (Fig. 2). Specifically, the phylogenetic analyses from the molecular and combined datasets consistently recovered the family Diogenidae *sensu lato* as a paraphyletic group, with two *Coenobita* species from Coenobitidae deeply nested within a subclade of the studied diogenid members with high support (89 bs, 0.99 Pp). The genus *Paguristes* was also found as non-monophyletic with *Paguristes doederleini* and *P. miyakei* clustering with the *Diogenes* species (97 bs, 1.00 Pp), while the remaining three *Paguristes* species (*P. albimaculatus*, *P. calvus* and *P. seminudus*) grouped together (100 bs, 1.00 Pp). The genera *Paguristes* and *Diogenes* clustered together with high support (81 bs, 0.97 Pp), being sister to the remaining diogenid genera. The genera *Dardanus*, *Coenobita*, *Ciliopagurus*, *Calcinus* and *Clibanarius* formed a statistically high supported clade (1.00 Pp, 100 bs). Within this clade, *Dardanus* was recovered as the earliest diverging genus followed by *Coenobita* belonging to the family Coenobitidae. On the other hand, the genera *Calcinus* and *Clibanarius* showed sister relationships, forming a cluster with *Ciliopagurus*. Although the Bayesian supports for the latter groups were relatively low (Pp < 0.85), relatively high bootstrap supports were detected in the ML analysis (bs > 70). Our combined (molecular + morphology) phylogeny recovered the relatively high

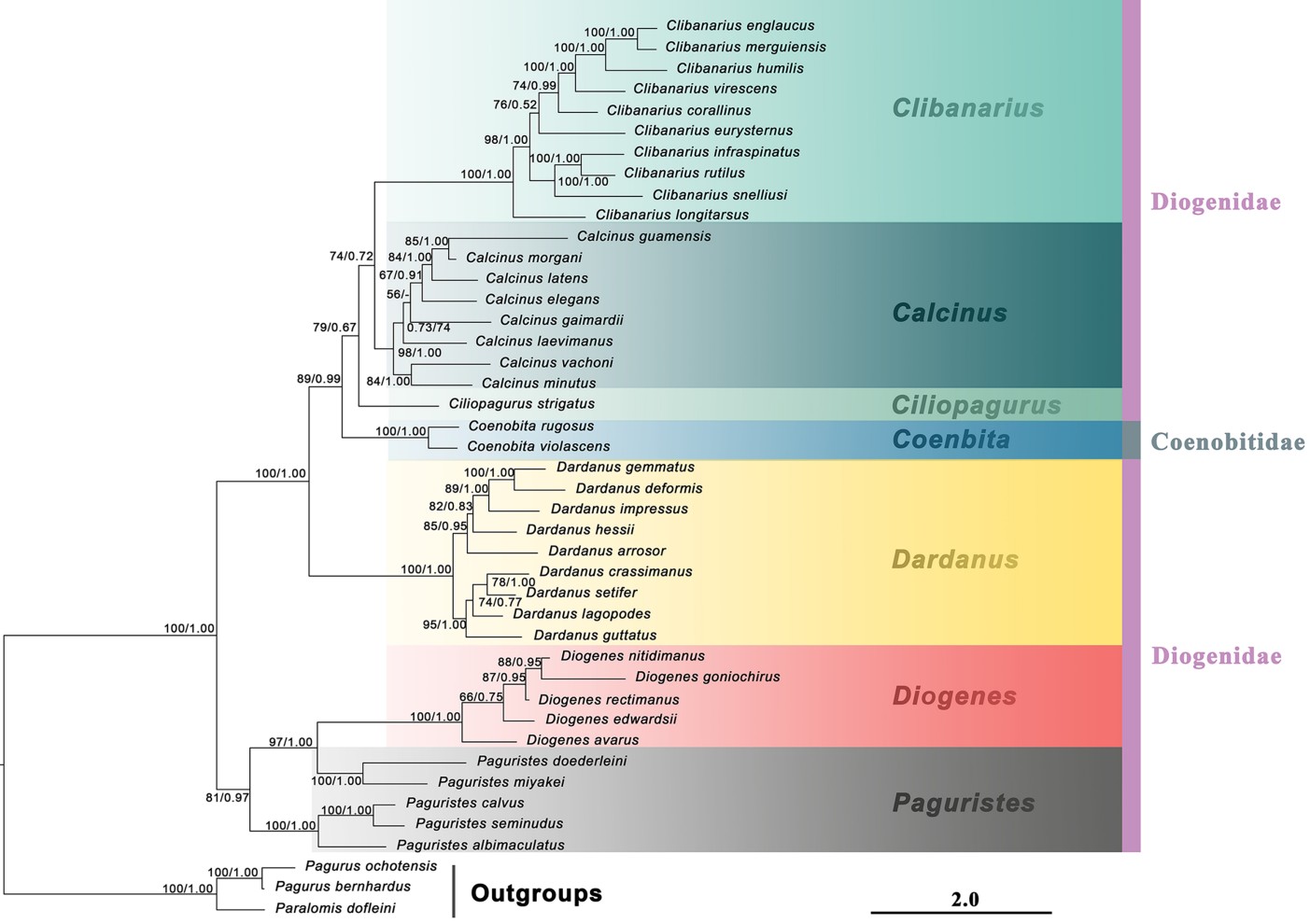

**Figure 1 Maximum likelihood phylogram based on two mitochondrial (16S rRNA and COI) and two nuclear (NaK and PEPCK) genes.** The studied diogenid genera are marked by different background colors. Vertical colored bars represent families, and the black vertical line represents outgroups. Nodal support values represent maximum likelihood bootstrap (bs)/Bayesian posterior probabilities (Pp). Only bootstrap support values above 50% are shown.                                                

Bayesian supports for these groupings (Fig. 2), which is taken as the best estimate for the phylogenetic relationships among the studied diogenid genera.

Maximum parsimony and maximum likelihood methods recovered approximately congruent ancestral state reconstructions for the three morphological characters (Fig. 3). The mosaic pattern of gill number in diogenid phylogeny appeared to suggest its phylogenetic insignificance for diogenid generic groupings (Fig. 3A). The carapace shield unarmed or with scattered setae appeared to be the derived state in the genera *Dardanus*, *Ciliopagurus*, *Calcinus* and *Clibanarius* (Fig. 3B). Within the *Paguristes-Diogenes* clade, hermit crabs with armed telson were genetically distinct from those with unarmed telson (Fig. 3C). In the case of another clade, however, hermit crabs with armed telson did not form a monophyletic grouping.

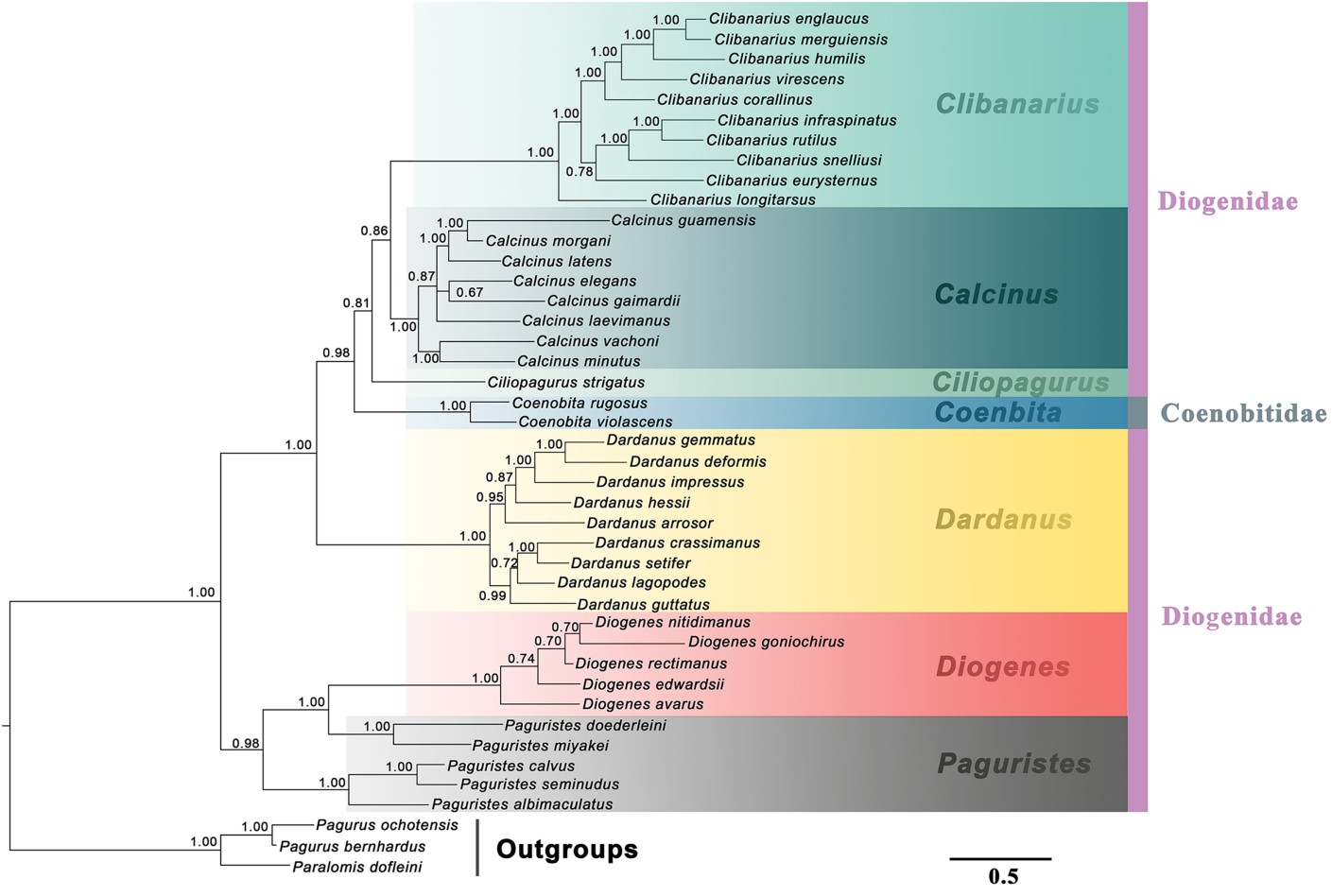

**Figure 2 Bayesian phylogram based on combined molecular (2,308 characters) and morphological (41 characters) data.** The studied diogenid genera are marked by different background colors. Vertical colored bars represent families, and the black vertical line represents outgroups. Bayesian posterior probabilities (Pp) are represented as percentages and >50% are noted above branches.    

## DISCUSSION

### Discordance between mtDNA and nDNA trees

Diogenidae *sensu lato* has been represented in phylogenetic studies of Anomura MacLeay, 1838 with the sample size of diogenids ranging from two (*Tsang et al., 2008*) to 23 taxa (*Bracken-Grissom et al., 2013*). Although these studies have enriched our understanding of diogenid relationships, dynamic diogenid classification and controversies over generic-level affinities emphasize the need for continued research of this group. In the present study, we combined 2,308 molecular (mitochondrial and nuclear) and 41 morphological characters from 38 taxa to recover the diogenid phylogeny. Surprisingly, we found conflicts between separate mtDNA and nDNA trees, which appears to illustrate the potential weakness of deriving diogenid phylogenies from mtDNA alone. It has been suggested that long branch attraction, which can confound phylogenetic analyses, might be more common among nodes deep in the tree (*Felsenstein, 2004*). Faster evolving genes (*e.g.*, mtDNA) likely exacerbate problems of long branch attraction, while slow-evolving
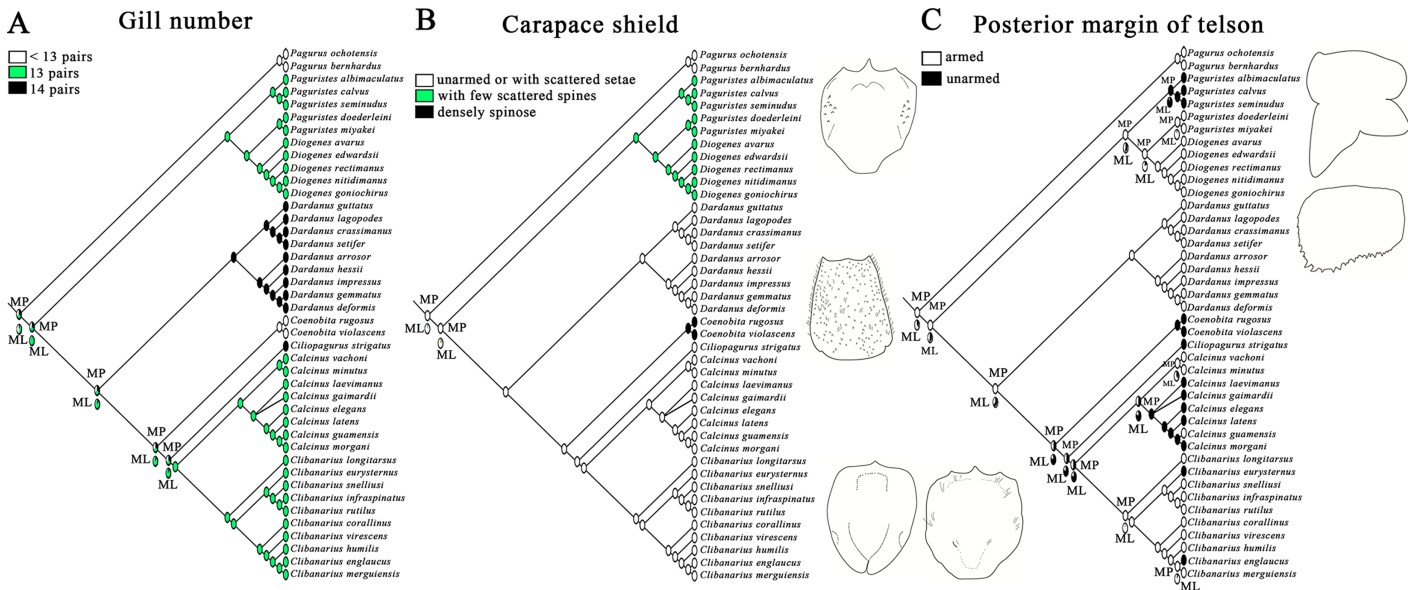

**Figure 3 Ancestral state reconstruction analysis using maximum parsimony and maximum likelihood methods for three morphological characters of gill number (A), carapace shield (B) and telson (C).** The proportional likelihoods of different ancestral states are indicated by the pie chart on the corresponding nodes. Line drawings of carapace shield and telson morphologies are made with a tablet (Wacom Intuos Pro PTH-851) by Dr. Yanrong Wang, and not to scale.

nDNA may help to resolve such problems (*Fisher-Reid & Wiens, 2011*). The potential of mtDNA and nDNA to resolve phylogenetic relationships at different levels (*e.g.*, deeper *vs.* more recent branches) has been considered as an important justification for obtaining and combining these data types for phylogenetic reconstruction in the first place (*Giannasi, Malhotra & Thorpe, 2001*; *Pereira, Baker & Wajntal, 2002*). This may be illustrated in the diogenid phylogeny by the usefulness of mtDNA dataset to resolve shallow branches (*Calcinus-Clibanarius-Ciliopagurus-Coenobita*) and nDNA to resolve relative deep branches (*Paguristes-Diogenes*). Our combined-data tree reflected the contributions of each dataset, and showed improved phylogenetic resolution and provided higher support for internal nodes. The addition of morphological data also increased the support for some intergeneric relationships that were poorly supported in the molecular-only phylogeny. These findings suggest that a combined evidence analysis is a useful approach for the elucidation of generic relationships of diogenid hermit crabs.

## Paraphyletic diogenidae with coenobitidae nested

As mentioned previously, some arguments exist for the monophyly of family Diogenidae *sensu lato*. Our molecular-based and combined (molecular + morphology) phylogenies consistently suggest paraphyly in Diogenidae *sensu lato* with the coenobitid genus *Coenobita* embedded within it, corroborating previous phylogenetic results (*Tudge, 1997*; *Schnabel, Ahyong & Maas, 2011*; *Tsang et al., 2011*; *Bracken-Grissom et al., 2013*; *Landschoff & Gouws, 2018*; *Craig & Felder, 2021*). As asymmetrical hermit crabs accompanied by an enlarged left chela, Diogenidae *sensu lato* and Coenobitidae are closely linked, which has been supported by somatically morphological grounds

(*McLaughlin, 1983*; *Martin & Abele, 1986*; *Richter & Scholtz, 1994*) as well as spermatozoa and spermatophore morphological evidence (*Tudge, 1997*), and recently supported by molecular phylogeny (*Schnabel, Ahyong & Maas, 2011*; *Bracken-Grissom et al., 2013*; *Craig & Felder, 2021*). Nevertheless, Coenobitidae represents a monophyletic taxon because the two coenobitid genera *Coenobita* and *Birgus* share an apomorphic type of $1^{st}$ antenna with specialized flagella and sensory hairs (*McLaughlin, 1983*; *Martin & Abele, 1986*), which is later confirmed by spermatological (*Tudge, 1997*) and molecular (*Bracken-Grissom et al., 2013*; *Craig & Felder, 2021*) phylogenies. In addition, a series of terrestrially-adapted apomorphies sets Coenobitidae apart from other paguroid families, including reduction in arthrobranch number, extreme reduction of the crista dentata and lateral compression of the ocular peduncles (*Bliss, 1968*; *Vannini, 1976*; *Mclaughlin et al., 2007*).

Altogether, the consistent placement of coenobitids within the family Diogenidae *sensu lato* revealed by our phylogenetic analyses, as well as those cited, indicates that taxonomic revisions are warranted for diogenids and coenobitids. One approach to revisions would be reassignment of coenobitids to the family Diogenidae *sensu lato* as a highly derived subfamily Coenobitinae (*Luque et al., 2017*). An alternative approach would be to subdivide the family Diogenidae *sensu lato* and reassign the existing genera to new families to reflect phylogenetic relationships. Given that Diogenidae *sensu lato* is a highly speciose and morphologically diverse family, it seems obvious that investigation of representatives from the missing diogenid genera is required to determine appropriate revisions at the family level.

## Generic-level relationships within diogenidae

The heterogeneity of the diogenid genera and the difficulty in determining their affinities have been previously suggested by *Forest (1984*, *1995)*. In this study, the investigated diogenid hermit crabs did not form a single clade but were split between two clades with strong nodal support, one grouping *Calcinus*, *Clibanarius*, *Ciliopagurus* and *Dardanus* and the other clustering *Paguristes* and *Diogenes* as sister genera. This contradicts morphological classification scheme for Diogenidae *sensu lato* proposed by *Fraaije, Van Bakel & Jagt (2017)*, in which *Calcinus*, *Ciliopagurus* and *Dardanus* was classified as members of a newly-erected family Calcinidae, and the other diogenid genera (including *Clibanarius*, *Paguristes* and *Diogenes*) constituted Diogenidae *sensu stricto*. Our results mirror those from the anomuran phylogenetic analyses (*Bracken-Grissom et al., 2013*) that included 23 diogenid hermit crabs with generic divisions consistent with our diogenid clades. However, intergeneric relationships in Diogenidae *sensu lato* have not been addressed in detail in their study. Additional support can be found in a recent molecular phylogeny of selected paguroid species from the western Atlantic (*Craig & Felder, 2021*), which revealed three groupings with generic compositions similar to those of our diogenid clades. Collectively, our results in combination with previous findings on the diogenid generic-level relationships concur in suggesting the necessity to revise the classification scheme of Diogenidae *sensu lato* based on fossil records (*Fraaije, Van Bakel & Jagt, 2017*).

The present phylogeny supports close relationships between *Calcinus*, *Clibanarius* and *Ciliopagurus*, and shows significant supports for a sister relationship between them and the

representatives of Coenobitidae (*Coenobita rugosus* and *C. violascens*), leaving representatives of the genus *Dardanus* at the base of this clade. Notably, *Ciliopagurus* is represented by only a single species, hence its placement should be treated with some level of caution. Ancestral State Reconstruction recovered carapace shield unarmed or with scattered setae as synapomorphy of diogenid genera in this clade. These results lend support to the diogenid generic groupings reported by *Craig & Felder (2021)* in which, however, little supports were recovered. The closer association between *Calcinus* and *Clibanarius* was also evident in the phylogeny of *Mantelatto et al. (2006)*, though their findings also included *Isocheles* Stimpson, 1858 and *Loxopagurus* Forest, 1964 in the clade. Morphologically, *Calcinus* and *Clibanarius* share the synapomorphies of 13 gill pairs, and larger chela palm and dactyl with tubercles. Ecologically, hermit crabs belonging to these two genera are usually found in intertidal or littoral habitats. These similarities in ecology are congruent with their close affinity in the phylogeny.

*Paguristes* is believed to be unique within Diogenidae *sensu lato* in having distinctive, paired first and second male pleopods modified as gonopods, non-cheliform fourth pereopods, and unpaired pleopods 3–5 occurring on the left side of abdomen only (*McLaughlin, 2003*; *Sha, Xiao & Wang, 2015*). However, *Paguristes* species selected in our study did not form a monophyletic group, which refutes the monophyletic hypothesis proposed by *Landschoff & Gouws (2018)*. As one of the most speciose genera of diogenid hermit crabs with diverse morphology and global distribution, *Paguristes* has been found closely allied with *Areopaguristes* Rahayu & McLaughlin, 2010 and *Pseudopaguristes* McLaughlin, 2002 morphologically and genetically (*Craig & Felder, 2021*), forming a taxonomically challenging group. Interestingly, a close link between *Paguristes* and *Diogenes* is supported by our total evidence approach, adding additional complexity to the classification of *Paguristes*. In agreement with *Craig & Felder (2021)*, the current study demonstrates a clear correlation between the shape of telson and genetic groupings of *Paguristes* and *Diogenes* species. Specifically, a subclade consisting of *Paguristes doederleini*, *P. miyakei* and *Diogenes* species corresponds to the morphologically defined group with armed telsons, while the remaining three *Paguristes* species (*P. albimaculatus*, *P. calvus* and *P. seminudus*) with unarmed telsons group together. These results further highlight the phylogenetic significance of telson morphology in revising relationships between *Paguristes* and their relatives (*McLaughlin & Provenzano, 1974*; *Craig & Felder, 2021*).

## CONCLUSIONS

The present results largely resolved the phylogenetic relationships of the studied diogenid genera, and revealed some salient problems in diogenid taxonomy. Diogenidae *sensu lato* is recovered as paraphyletic on the basis of an internally nested Coenobitidae, and a similar situation may occur for the genus *Paguristes* with *Diogenes* species deeply nested inside it. Considering the high species richness and morphological diversity of *Paguristes*, there is no doubt that more extensive analyses including deep taxon sampling are needed to progress knowledge of this taxonomically contentious genus. Nevertheless, *Paguristes* is well separated from an assemblage including *Calcinus*, *Clibanarius*, *Ciliopagurus* and *Dardanus*. This finding in combination with previous phylogenetic results concurs in

suggesting the evolutionary independence of *Paguristes*-like species. Future integrated studies with wide taxonomic coverage and multiple data sources are necessary to clarify the taxonomy and evolution of diogenid hermit crabs.

## ACKNOWLEDGEMENTS

We would like to express our sincere thanks to Prof. Tin-Yam Chan of National Taiwan Ocean University for the help in sample collection and advice on morphological identification.

### Funding

This work was supported by the National Key R&D Program of China (No. 2022YFC3102403), the National Science Foundation for Distinguished Young Scholars (No. 42025603), the Strategic Priority Research Program of the Chinese Academy of Sciences (No. XDB42000000), and the Qingdao New Energy Shandong Laboratory Open Project (No. QNESL OP202306). The funders had no role in study design, data collection and analysis, decision to publish, or preparation of the manuscript.

### Grant Disclosures

The following grant information was disclosed by the authors:
National Key R&D Program of China: 2022YFC3102403.
National Science Foundation for Distinguished Young Scholars: 42025603.
Strategic Priority Research Program of the Chinese Academy of Sciences: XDB42000000.
Qingdao New Energy Shandong Laboratory Open Project: QNESL OP202306.

### Competing Interests

The authors declare that they have no competing interests.

### Author Contributions

- Jiao Cheng conceived and designed the experiments, analyzed the data, prepared figures and/or tables, authored or reviewed drafts of the article, and approved the final draft.
- Wenjie Li performed the experiments, analyzed the data, prepared figures and/or tables, authored or reviewed drafts of the article, and approved the final draft.
- Yanrong Wang analyzed the data, prepared figures and/or tables, and approved the final draft.
- Zhongli Sha conceived and designed the experiments, authored or reviewed drafts of the article, and approved the final draft.

### Data Availability

   All sequences obtained in the present study are available at GenBank:
MK610008–MK610036; MK610039–MK610041; MH508045–MH508054;

MH508056–MH508087; MK747765–MK747780; MK747782–MK747788; MK747790–MK747812; MW411962; MW411963; MW413814; MW413815.

## Supplemental Information

Supplemental information for this article can be found online at http://dx.doi.org/10.7717/peerj.17922#supplemental-information.

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
