# Peer review of "Phylogenetic synthesis of morphological and molecular data reveals insights on the classification of diogenid hermit crabs (Crustacea: Decapoda: Anomura)"

_PeerJ, doi:10.7717/peerj.17922_

## Round 0.1 · original submission · Major Revisions

· Academic Editor

Major Revisions

Please follow the two reviewers' comments to improve your manuscript.

**Language Note:** The review process has identified that the English language must be improved. PeerJ can provide language editing services - please contact us at [email protected] for pricing (be sure to provide your manuscript number and title). Alternatively, you should make your own arrangements to improve the language quality and provide details in your response letter. – PeerJ Staff

Reviewer 1 ·

Basic reporting

.

Experimental design

.

Validity of the findings

.

Additional comments

In general, this manuscript titled as” More than meets the eye: Phylogenetic synthesis of morphological and molecular data reveals insights on the classification of diogenid hermit crabs (Crustacea: Decapoda: Anomura)” reports the results of a combined molecular and morphometric phylogeny of Diogenid hermit crabs. Due to the scarcity of molecular data of this group, the presented result will provide important information to understand the molecular systematics of Anomuran. Nonetheless, several important revisions need to be made before this manuscript can be published.

General comments
The overall manuscript should be proofread by a fluent English speaker. It is very difficult to stay focused.

Title: Change the title to a simpler but informative form. This type of title may fit for conference or symposium presentation.

Abstract: Rewrite the abstract to clearly understand the purpose of the study. Especially mentioning the sample size (38 species?), and the number of molecular data for each gene, as the authors mentioned 41 Morph-characters. And at Line 39-40, is that statement a conclusion for Diogenus group? I mean the authors tasted 41 Morph-characters, among them only these two are effective for identifying this genus. If so, emphasize it as a diagnostic character for this group.

Introduction: Rewrote it with focusing on the purpose of the present study.



Materials and Methods: The overall part is very vague. Please write sequentially and simply.

Taxon sampling and data collection: Clearly state the number of samplings for each gene or morphometric characters. Even though it is provided in Table 1 and in Supp. Table, respectively; however, state it here clearly.

DNA extraction, PCR amplification and sequencing
Line 140: according to the manufacturer protocol or the authors’ own. Mention it or, for the latter case, provide details. Because getting plenty amount of DNA is not easy from hermit crabs.
Lines 143-147: Provide all the used primer sequence compiling in a table with reference.
Line 171: molecular dataset (provide the used number of nt in analyses, 2308?)
In Line 178 “(morphology + molecular) datasets”, and in Line 221 “(molecular + morphology) phylogeny”. Which one? Keep consistency.
Line 198: “of three morphological characters of…... “. The authors tested 41 characters. What is the rational of selecting only three among the 41. Better to mention here to reduce confusion for the reader.

Result and Discussion: Rewrite to keep the order.

Reviewer 2 ·

Basic reporting

The authors examined in detail the morphological and molecular data in the classification of hermit crabs. Using two mtDNA gene regions and two nuclear DNA gene regions, they examined thirty-eight species belonging to six genera and supported these data with forty-one morphological characters to estimate the largest phylogeny of hermid crabs to date. As a result of the analyses, it was concluded that taxonomic revision was needed in some genera (e.g. Paguristes).

Experimental design

no comment

Validity of the findings

no comment

Additional comments

Authors can check the minor corrections I have made in the text.

Annotated reviews are not available for download in order to protect the identity of reviewers who chose to remain anonymous.

---

## Round 0.2 · Minor Revisions

· Academic Editor

Minor Revisions

Fear authors,

Please follow the remaining reviewer's minor comments and make the changes you agree, if you don't agree with the reviewers' comments explain why.

Reviewer 2 ·

Basic reporting

The article has been reorganised according to the suggested corrections and major errors have been corrected. The article is well-written in clear, concise English and reports all results meaningfully and clearly. I found it an easy and enjoyable read. The article highlights the importance of morphological and molecular data in the classification of hermit crabs.

Experimental design

no comment'

Validity of the findings

no comment'

Additional comments

no comment'

Reviewer 3 ·

Basic reporting

This manuscript is well structured and written but English language could be improved.

The present study sheds light on the previously controversial generic phylogeny of Diogenidae and highlights the need for detailed taxonomic revisions of this family as well as some genera (e.g., Paguristes) to reconcile current classifcations with phylogenetic relationships. Thus, an improved understanding of this group would shed light on not only its diversity and ecology but also an evolutionary history of decapod crustaceans.

In the part Introduction the authors made a comprehensive review of the morphological and molecular data available for this group. The text could to be revised to be mole clear.
The part Material and methods was comprehensive written with detailed information for the methodology applied. The Results are clearly presented and illustrated with appropriate Figures and Tables. The Tables and Figures give useful information, which verify the analyzed data and conclusions.

Experimental design

no comment

Validity of the findings

no comment

Additional comments

no comment

---

## Round 0.3 · accepted · Accept

· Academic Editor

Accept

The authors have addressed all of the reviewers' comments and that I am happy with the current version. therefore, this manuscript is ready for publication.